# Effects of Feedback-Supported Online Training during the Coronavirus Lockdown on Posture in Children and Adolescents

**DOI:** 10.3390/jfmk7040088

**Published:** 2022-10-14

**Authors:** Oliver Ludwig, Carlo Dindorf, Torsten Schuh, Thomas Haab, Johannes Marchetti, Michael Fröhlich

**Affiliations:** 1Department of Sports Science, Technische Universität Kaiserslautern, 67663 Kaiserslautern, Germany; 2Sport Performance Education, 66386 Sankt Ingbert, Germany; 3Institute of Sports Science, Saarland University, 66123 Saarbrücken, Germany

**Keywords:** posture training, feedback, COVID-19, neck-shoulder-region, shoulder protraction, upper crossed syndrome, posture weakness, physical inactivity, sedentary behavior, soccer

## Abstract

(1) Background. The coronavirus pandemic had a serious impact on the everyday life of children and young people with sometimes drastic effects on daily physical activity time that could have led to posture imbalances. The aim of the study was to examine whether a six-week, feedback-supported online training programme could improve posture parameters in young soccer players. (2) Methods. Data of 170 adolescent soccer players (age 15.6 ± 1.6 years) were analyzed. A total of 86 soccer players of a youth academy participated in an online training program that included eight exercises twice per week for 45 min (Zoom group). The participants’ exercise execution could be monitored and corrected via smartphone or laptop camera. Before and after the training intervention, participants’ posture was assessed using photographic analysis. The changes of relevant posture parameters (perpendicular positions of ear, shoulder and hips, pelvic tilt, trunk tilt and sacral angle) were statistically tested by robust mixed ANOVA using trimmed means. Postural parameters were also assessed post hoc at 8-week intervals in a control group of 84 participants of the same age. (3) Results. We found a statistically significant interaction (*p* < 0.05) between time and group for trunk tilt, head and shoulder protrusion and for hip anteversion in the Zoom group. No changes were found for these parameters in the control group. For pelvic tilt no significant changes were found. (4) Conclusions. Feedback-based online training with two 45 min sessions per week can improve postural parameters in adolescent soccer players over a period of six weeks.

## 1. Introduction

Postural disorders are widespread in childhood and adolescence and are reported with a prevalence of 25–60% [1,2,3]. A protracted shoulder girdle, a protracted head, a hollow back, and a hunchback are the most common postural weaknesses in adolescents [4]. Current studies have shown that this can lead to pain at an older age. For example, Dolphens et al. demonstrated in 1196 children and adolescents that increased forward head tilt was associated with lifetime prevalence of neck pain [5]. Kim et al. in turn were able to demonstrate, that a protracted head could lower respiratory functions in young adults [6]. An important cause for the occurrence of postural weaknesses is apparently muscular imbalances that lead to the displacement of individual body parts [7,8]. A sedentary everyday life seems to promote such imbalances, and the intensive use of smartphones also apparently leads to postural adaptations, especially forward displacement of the shoulder and head [9,10].

In 2020, the coronavirus pandemic had a serious impact on the everyday life of children and young people through measures such as lockdowns and temporary curfews. In Germany, as in many other countries around the world, young people who were normally active in sports were cut off from outdoor exercise and sports activities in clubs for several months. While physical activity time thus decreased significantly, time in front of the computer increased—as a leisure activity, but also because school lessons were largely held online. This promoted physical inactivity among many adolescents in Germany and worldwide for weeks, with sometimes drastic effects on daily physical activity time. For example, Schmidt et al. showed that in Germany, among 14–17-year-old boys, physical activity decreased by an average of 21 min per day during Corona, while time spent in front of the screen relaxing increased by 79 min per day [11]. A survey of over 6400 children and adolescents in France by Chambonniere et al. also showed that 57% of teenagers significantly decreased their physical activity during Corona, which was particularly evident in adolescents who were previously more active in sports [12]. Guan and colleagues found the same trend in Canadian adolescents, namely, in addition to a decrease in physical activity time, a 66% increase in time spent watching television and a 35% increase in time spent playing video games [13].

It is well known that there is a strong association between health outcomes and physical activity behavior. Guan et al. therefore concluded that the health of children and adolescents may have been compromised due to a lack of physical activity during the Corona pandemic [13].

Negative effects of sitting on altered muscular activity and misalignments of the head and neck have been demonstrated. For example, Caneiro et al. found the typical postural deficiencies in the form of increased neck flexion and increased anterior translation of the head as a result of a slump sitting position, which led to an unfavorable increase in the activity of the postural muscles in the neck region [14], while Ertekin and Günaydin demonstrated a change in muscle activity and stiffness of the trapezius muscle associated with protracted shoulders [15].

Many of these muscularly induced misalignments are associated with neck pain. For example, Ruivo et al. observed protracted head in 68% of the subjects and protracted shoulders in 58% of the subjects in studies of 275 adolescents, and forward head posture was significantly associated with neck pain [16].

According to current scientific knowledge, an important starting point in the prevention of shoulder and neck pain is the strengthening of the stabilizing musculature [17]. Targeted sports activities can contribute to this.

One approach during the pandemic time, which was physically and psychologically stressful for children and adolescents, was sports activities carried out online in groups. This is where the present study comes in. For adolescent soccer players at a DFB (German Football Association) youth academy, who normally trained together three to four times a week on the sports field, posture-specific sports training was offered online twice a week via the Zoom^®^ platform during the lockdown phase. The typical postural weaknesses of this target group were known from previous own studies [8], so the training programme was adapted accordingly. As the online meetings took place live via smartphone or laptop cameras, the exercises could be individually corrected by trainers in real time for each participant.

Although some studies discuss the general use of digital tools and online training or effectiveness regarding improvements in overall activity levels during the coronavirus pandemic, there is insufficient research on the effectiveness of online training on body posture. Tjønndal was able to show that the use of online tools during the corona pandemic was quite uncoordinated and that synchronized (live-streamed) training was offered rather rarely [18]. On the other hand, when Parker et al. surveyed 963 adolescents, they found that the willingness to use digital platforms to support and guide physical activity was present in 26.5% of the surveyed adolescents [19]. These are initially good conditions, especially since 95–96% of 13–18-year-olds in Germany had their own smartphone at the end of 2021 [20].

However, there is very little research on the use and effectiveness of synchronized (live streamed) online training, particularly for posture correction and especially when participants received live feedback. Given the potential and benefits in situations where face-to-face training cannot take place, there is a clear need for research. Therefore, the aim of the study was to examine whether a six-week, feedback-supported online training programme could improve posture parameters in children and adolescents.

## 2. Materials and Methods

The online training intervention took place during the second “hard” coronavirus lockdown in Germany, which came into effect on 16 December 2020 and lasted for six weeks.

### 2.1. Subjects

A total of 96 adolescents from the U14 (age 13–14 years) to U19 (age 17–19 years) youth teams of the DFB youth academy of the SV 07 Elversberg soccer club (Germany) participated in the training intervention (Zoom group). Participation was voluntary and was advertised via the club. The data of 86 participants of the Zoom group were used for final analysis (anthropometric data in Table 1); data of ten subjects had to be removed because regular participation was not possible (due to poor internet connection or illness) or the follow-up appointment was missed.

Since, for ethical reasons, we could not deprive young soccer players of the club of participating in online training during the coronavirus lockdown, no non-training control group could run in parallel. We therefore selected an age-matched control group of 84 adolescent soccer players post hoc, who were examined twice at intervals of eight weeks and only took part in regular soccer training or had a training break in between.

All participants or their legal representatives were fully informed beforehand and gave their written informed consent to participate in the postural analyses and to take part in the video conferences. The study was conducted in accordance with the Declaration of Helsinki and approved by the local University Ethics Committee (ref. nr. 6-18).

### 2.2. Posture Analysis

Before the start of the study, a posture analysis was carried out on all participants on the premises of the association. Due to the contact restrictions, an exemption was obtained from the responsible municipal regulatory authority. In view of the current infection situation, appropriate corona precaution measures were followed strictly.

During the analysis, the subjects were first weighed in their underwear, and their height was determined (SECA stadiometer). The following anatomical landmarks were marked with marker dots or marker balls (diameter 12 mm): malleolus lateralis, greater trochanter, acromion, spinous process of the 7th cervical vertebra (C7), spinous process of the 1st sacral vertebra (S1), anterior superior iliac spine (ASIS), and posterior superior iliac spine (PSIS). The subjects then placed themselves in front of a measuring wall and assumed a relaxed posture (arms hanging loosely, looking straight ahead, breathing normally). A posture photo was taken using a camera (Logitech webcam, Full HD) mounted on a tripod at hip height. The inclination of the sacrum was then determined in degrees by placing an electronic inclinometer (Neoteck NTK033) on the proximal part of the sacrum.

Using analysis software (Dartfish ProSuite 6, Dartfish, Fribourg, Switzerland), the following posture parameters were determined (Figure 1): horizontal distances of the ear, shoulder (acromion), and hip (major trochanter) perpendicularly through the lateral malleolus as a percentage of body height; pelvic tilt (angle between the connecting the PSIS–ASIS line to the horizontal); and trunk forward tilt (angle between the connecting the S1–C7 line to the vertical).

Posture assessment using analysis of angles and distances from posture photographs is a scientifically well-studied, reliable and valid procedure [21,22].

### 2.3. Intervention

To implement the intervention, a temporary film studio was set up in the club’s athletics hall, from which the training exercises were broadcast live twice a week via the Zoom^®^ (Zoom Video Communications Inc., San José, CA, USA) communications platform. One trainer performed the exercises at the designated pace, while a second trainer commented on the execution of the exercises and announced the number of repetitions and rest times. The participants had previously been asked to point their smartphone or laptop camera so that they were clearly visible in the front or side profile. Two other trainers switched from participant to participant via video during the exercise session and controlled and, where necessary, verbally corrected the execution of the exercise until the participant executed it correctly (Figure 2). The participants were divided into a total of three subgroups of 30–35 adolescents, who took part in the online training at different times to ensure sufficient monitoring by the trainers. The first two sessions were necessary to learn how to perform the exercises correctly.

From the postural examinations (pre-test), it was known that there were deficits especially in the head and shoulder area (protracted shoulder girdle, head protrusion), as well as in the pelvic area (Table 2). Therefore, the exercises were chosen to meet the following criteria:

1.Strengthening of the target muscles for segmental postural improvement;2.Feasibility with aids available in the household;3.Easy to correct via video.

To improve the position of the shoulder and head, particular emphasis was placed on strengthening the rhomboid, latissimus dorsi, rectus capitis, longus colli, longus capitis, and obliquus capitis muscles [23]. To correct the pelvic position in the sense of reducing the pelvic tilt, the rectus abdominis, gluteus maximus, and ischiocrural muscle groups were strengthened [24]. The exercises performed are described in Table 3.

The exercise sessions were performed twice a week in the afternoon, three days apart, and lasted about 45 min each. At the beginning, the participants completed an approximately five-minute warm-up programme consisting of running on the spot, jumping jacks, knee bends, and light stretching exercises for the trunk and leg muscles. This was followed by the eight exercises with 20 repetitions each (Table 3). This sequence was repeated a total of three times with short drinking breaks. The training was concluded with five minutes of stretching exercises for the trunk and legs and a check of the load intensity.

The exercises were demonstrated by a trainer in real time and, thus, the number of repetitions, load, and rest times were specified. An adjustment to the individual performance level of the participants was made through the choice of weights (e.g., water bottles of different sizes), the height of the obstacles (for the “swimmer” exercise), and the tension of the resistance band (e.g., for rowing exercises). In the first week of training, the correct execution of the movement was trained and the weights were adjusted so that the number of repetitions could be carried out. After each training session, the participants were asked to rate the training load on a ten-point Likert scale via chat. The target range was set at 6–8; if the target was exceeded or not reached, the resistance and weights were varied individually in consultation with the participants. After seven training sessions, either the weights or the number of repetitions (now 30) was increased. This was maintained until the 12th training session.

### 2.4. Statistics

Outliers at the time of the pre-tests were checked again in the posture photos. As these were not measurement errors but actual extreme values in the range of the posture parameters, they were not removed.

Due to presence of outliers, as well as partial inhomogeneity of the covariance according to Box’s test (*p* < 0.01), assumptions could not be met for the calculation of a commonly used mixed ANOVA. Therefore, a robust mixed ANOVA using trimmed means was applied for statistical analysis of the interaction effects of the groups and measuring points in time using the R package WRS2 [25]. Further post hoc statistical analysis was performed using IBS SPSS Statistics (version 16, SPSS Inc., Chicago, IL, USA). Visualizations were performed using the Python library “Seaborn” [26].

## 3. Results

Table 4 shows the development of the posture parameters in the training and control groups, while Figure 3 shows the distribution and development of the parameters over time.

### 3.1. Interaction Effects

There was a statistically significant interaction between time and group for head protrusion (F(1, 93.75) = 51.92, *p* < 0.001, partial η^2^ = 0.36), for shoulder protrusion (F(1, 93.85) = 23.62, *p* < 0.001, partial η^2^ = 0.20), and for hip anteversion (F(1, 100.99) = 7.07, *p* = 0.009, partial η^2^ = 0.07).

We found a statistically significant interaction between time and group for trunk tilt (F(1, 96.03) = 40.65, *p* < 0.001, partial η^2^ = 0.30).

For pelvic tilt, we found no statistically significant interaction between time and group (F(1, 99.43) = 1.93, *p* = 0.168). Further, no main effects for time (F(1, 92.49) = 0.15, *p* = 0.695) or group (F(1, 99.43) = 0.48, *p* = 0.492) could be found.

There was no statistically significant interaction between time and group for sacral angle (F(1, 100.99) = 2.44, *p* < 0.001). There were, however, main effects for time (F(1, 97.96) = 28.64, *p* < 0.001, partial η^2^ = 0.23) and group (F(1, 100.99) = 48.94, *p* < 0.001, partial η^2^ = 0.33).

### 3.2. Main Effects of “Between-Subject Factors” (Group)

The control and training groups did not differ significantly at the pre-test on the variables head protrusion, shoulder protrusion, hip anteversion, and pelvic tilt (*p* > 0.05), but there were differences in trunk tilt and sacral angle (*p* < 0.05). At the post-test, the training and control groups differed significantly (*p* < 0.05) in all the variables except pelvic tilt (*p* > 0.05).

### 3.3. Main Effects of “Within-Subject Factors” (Time)

For the training group, there was a significant change between the pre- and post-tests for all parameters (*p* < 0.05), except for pelvic tilt. For the control group, there was no difference between the measurement times (*p* > 0.05), except for the variables sacral angle and trunk tilt (*p* < 0.05).

## 4. Discussion

The coronavirus pandemic brought drastic changes to the everyday lives of many people, especially during the phases of lockdown, with contact restrictions and massive reductions in social and sporting life. Children and adolescents were particularly affected [11]. The subjects of our study were young soccer players in the junior performance sector who previously trained three to four times a week for 90 min or more with an additional game at the weekend. During the phases of the second hard lockdown, which lasted from mid-December 2020 to May 2021 in Germany, club life was largely switched off. Training sessions conducted at home and accompanied online were therefore the only guided sporting activity for many young people and, at the same time, one of the few opportunities to maintain social contacts within the team.

The present study was able to show that significant improvements in posture could be brought about by the six-week online training programme. To substantiate the study results, the changes in posture parameters during the six weeks of training were compared post hoc with the posture development of a roughly identically composed group of young competitive soccer players in regular sporting activity after the coronavirus lockdown phase. Although a control group running at the same time would have been scientifically optimal, it could not be realised at the time the study was conducted for ethical reasons, as we did not want to deny any young person participation in the exercise programme during the lockdown. The absence of systematic differences between both groups, and, therefore, the potential suitability, is also indicated by the mostly non-significant differences for the pre-test condition.

The reduction in the perpendicular distances of the ear, the shoulder, and the hip show that an improvement in posture in the sense of a less forward-leaning position could be achieved, presumably through the training-related strengthening of the dorsal muscle chain. Figure 4 shows a typical example of the change in a participant’s habitual posture over the course of the study. It is known that deviations of the head and trunk from the perpendicular can be associated with pain in the neck and back [27]. The violin plots in Figure 4 provide information about the distribution of these parameters and show that there was no normal distribution in the pre-tests, but rather a skewed distribution in the direction of the perpendicular. Minimum perpendicular distances are considered biomechanically optimal.

As a measure against protracted head, the longus colli, obliquus capitis, and longus capitis muscles were strengthened via Exercises 1 and 8 (Table 3). For retraction of the shoulders, it was additionally and synergistically important to strengthen the rhomboid, latissimus dorsi, and trapezius muscles, which were activated in Exercises 1, 2, and 5. This is consistent with the findings of Ruivo et al. [28], who were able to improve head and shoulder position in adolescents through a 16-week strength and stretching programme, and also with the work of Sheikhhoseini et al. [29], who were able to demonstrate the effect of appropriate interventions on head position.

Even though the training group and the control group differed in the parameter trunk tilt at the time of the pre-test, an improvement was observed in the training group in the post-test, but with even a deterioration in the control group. To reduce trunk tilt, the dorsal muscle chain (especially the gluteus maximus, erector spinae, and biceps femoris muscles) had to be strengthened (Exercises 5, 6, 7 in Table 3). The work of Dolphens et al. [5,30] showed that a forward-leaning posture in adolescence is an important predictor of back pain in the following years.

In the area of pelvic alignment, we could not find any significant improvements in the intervention group. A lifting of the anterior pelvic edge (reduction in the pelvic tilt and decrease in the sacral angle) would be associated with a reduction in lumbar lordosis after a new muscular balance between the inserting muscle groups has been established [7]. Strengthening the muscles that straighten the pelvis (especially the rectus abdominis, gluteus maximus, and hamstrings) would have been responsible for this. The training programme specifically included exercises for this purpose (1, 3, 4, 6, 7; Table 3). The fact that the training intervention did not bring about any improvement in this area could be related to the fact that a change in the pelvic position in the resting posture can presumably be brought about less through an increase in the strength of the muscles involved and more through an improvement in the proprioceptive perception of the pelvic position [24,31]. However, proprioceptive exercises were not part of the training programme. Furthermore, it cannot be ruled out that the training stimuli on the muscle groups mentioned were too low.

The control group did not show any improvement in the posture parameters (except for the sacral angle, which also decreased in the training group), so we assume that the online training was causally responsible for the improvement in the training group. This seems relevant to us insofar as the positive effects of targeted strength training on posture, which have already been proven in other studies [24,32,33,34], could also be achieved via online feedback training.

We see a significant aspect that contributed to the success and motivation of the participants in the feedback we were able to give in real time via video. It has been shown that in this way, frequent mistakes could be corrected, which otherwise would probably have led to a lower training success. At the same time, this kind of contact was an important psychological factor for many young people, as they reported afterwards. Praise and motivational announcements helped many participants to persevere through the strenuous training sessions and to maintain their bond with the team during the coronavirus period. From a scientific point of view, the lockdown situation naturally also had the advantage that (apart from individual sport at home) there were few outside influences in sporting terms, i.e., there was almost a “laboratory situation”.

We see online feedback as a great opportunity for internet training, which is clearly superior to traditional training videos that are available in large numbers via social media and platforms such as YouTube^®^ but offer no feedback option [18].

Our study has several limitations. As already mentioned, the intervention and control groups could not run in parallel. Nevertheless, the control group was composed as homogeneously as possible to the training group, even though they underwent regular sports training. However, since posture-relevant exercises were not part of their soccer training, we can exclude strong influencing effects here. It must also be emphasised that, apart from the load control, the training programme could not be individually adapted. Participants without postural deficits also performed the same exercises, so that a smaller effect of the intervention could be assumed here, which, in the end, rather led to an underestimation of the training effect. Nevertheless, training the postural muscles is important and makes sense in these cases as well, since many of the trained muscle groups also play a major role in movement stabilisation in soccer. In the selection of exercises, the choice of tools was limited to those available to the participants at home (water bottles as a substitute for dumbbells, exercise balls, resistance bands, broomsticks). Even though the adolescents had the specification not to do any additional training at home, this was not a factor that could be realistically controlled, but would not diminish the basic statement.

Our study was limited to performance-oriented youth soccer players. Nevertheless, we assume that the approach of feedback-supported training could also be successful for other youth target groups.

## 5. Conclusions

In the context of our study, it has been shown that it is possible to provide individual exercise supervision and correction for participants during online posture training. Feedback-based online training with two 45 min sessions per week can improve postural parameters in adolescent soccer players over a period of six weeks.

## Figures and Tables

**Figure 1 jfmk-07-00088-f001:**
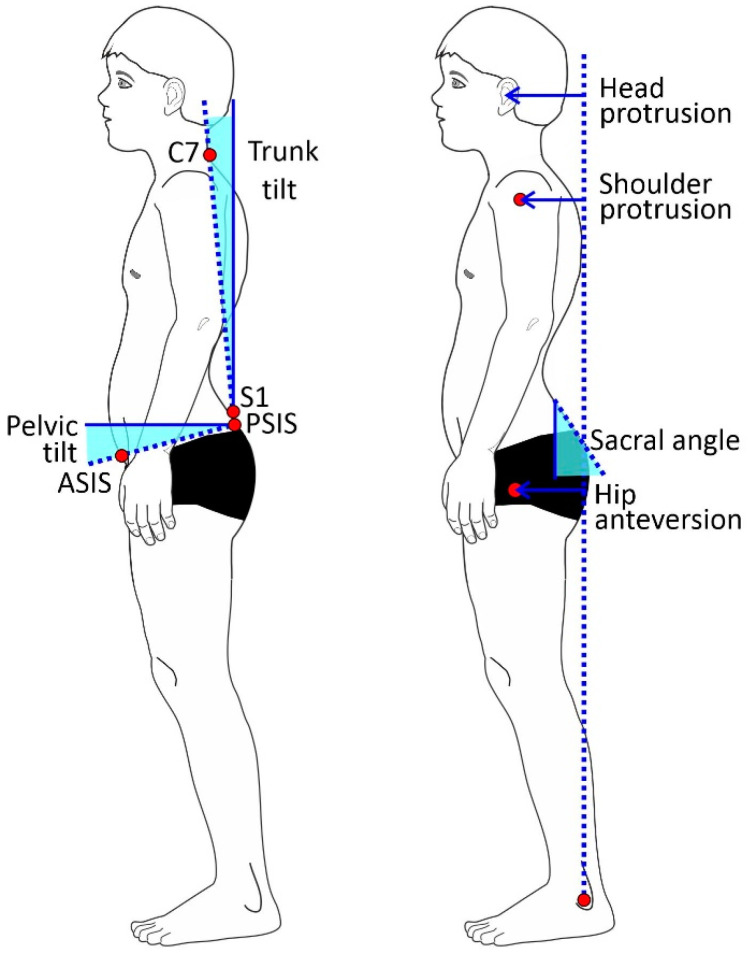
Analyzed posture parameters; ASIS = anterior superior iliac spine, PSIS = posterior superior iliac spine, C7 = 7th cervical vertebrae, S1 = 1st sacral vertebrae.

**Figure 2 jfmk-07-00088-f002:**
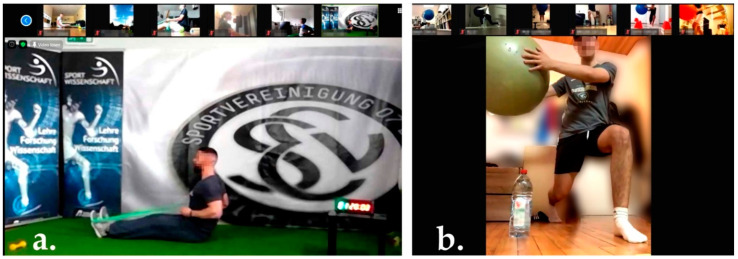
Exemplary online view that the participants (**a**) and the correcting trainers (**b**) had during the exercise sessions. In the photo on the right, the background has been changed for data protection reasons. The persons depicted have given their written consent to the publication of the photos.

**Figure 3 jfmk-07-00088-f003:**
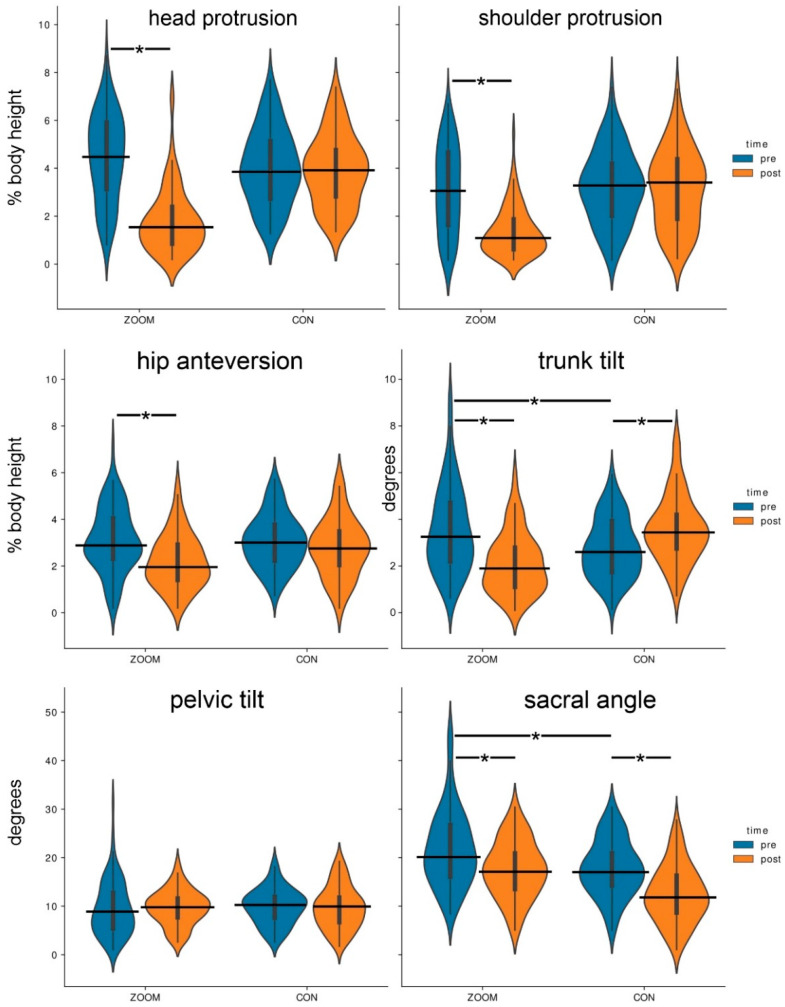
Violin plots of the development of the posture parameters of the training group (ZOOM) and the control group (CON). The horizontal lines mark the median, * = significant difference (*p* < 0.05).

**Figure 4 jfmk-07-00088-f004:**
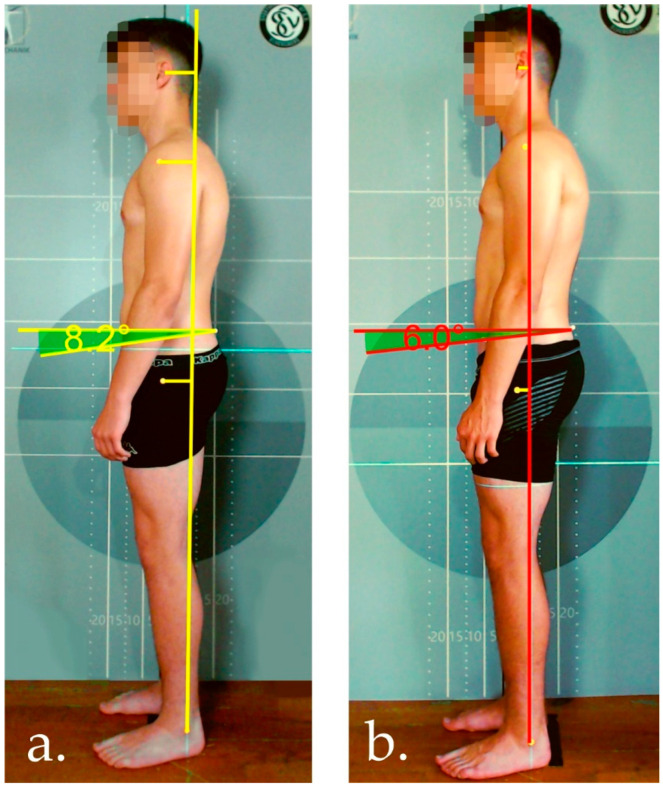
Typical example of the change in a participant’s habitual posture (**a**: pre-test; **b**: post-test).

**Table 1 jfmk-07-00088-t001:** Anthropometric data of the subjects.

	Zoom (N = 86)	Control (N = 84)
Age [years]	15.6 ± 1.6	15.7 ± 1.6
Weight [kg]	63.0 ± 12.2	65.3 ± 10.9
Height [cm]	173.0 ± 9.8	174.1 ± 9.0

**Table 2 jfmk-07-00088-t002:** Prevalence of postural deviations in the studied population (absolute N/prevalence).

Age	N	Forward Head	Protracted Shoulder	Scapulae Alatae	Pelvic Anteversion	Increased Pelvic Tilt	Hunchback
13–14	17	8/47%	10/59%	3/18%	2/12%	8/47%	2/12%
14–15	17	11/65%	12/71%	1/6%	11/65%	3/18%	7/41%
15–16	20	11/55%	11/55%	3/15%	11/55%	4/20%	2/10%
16–17	20	8/40%	7/35%	2/10%	10/50%	1/5%	5/25%
17–19	22	14/64%	11/50%	1/5%	15/68%	3/14%	5/23%
Sum	96	52/54%	51/53%	10/10%	49/51%	19/20%	21/22%

**Table 3 jfmk-07-00088-t003:** Description of the exercises and the targeted muscle groups.

Exercise	Target Muscles	Execution
#1 Swimmer	- M. deltoideus pars spinalis - M. rhomboideus major and minor - M. trapezius pars ascendens - M. erector spinae - M. gluteus maximus - M. biceps femoris - M. longus colli - M. longus capitis	*Starting position:* Prone, arms and legs extended, feet slightly off the floor, head slightly raised, looking down towards the floor. *Execution:* Bring arms back in a motion similar to breaststroke with the thumb pointing upwards. Raise the arms above bottles placed at the sides at shoulder height and then lower them again. Consciously pull the shoulder blades together. Keep the legs in a straight position.
#2 Reverse Butterfly in Standing Position	- M. deltoideus pars spinalis - M. trapezius pars ascendens - M. rhomboideus major and minor	*Starting position:* Stand hip-width apart, knees slightly bent, upper body bent forward. Keep back straight throughout the exercise. Hold dumbbells/bottles in upper grip and bend arms slightly. *Execution:* Raise arms sideways until elbows are level with shoulders, contracting scapulae. Slowly lower the arms to the sides.
#3 Torso Rotation with Lunge Backward	- M. obliquus internus abdominis - M. obliquus externus abdominis - M. transversus abdominis - M. deltoideus pars clavicularis	*Starting position:* Lunge backwards, hold exercise ball in front of the body with arms outstretched, tense abdomen. *Execution:* Slowly rotate the upper body as far as possible, always looking at the ball, hold briefly and then rotate to the other side.
#4 Push-Ups on the Ball	- M. rectus abdominis - M. transversus abdominis - M. obliquus internus abdominis - M. obliquus externus abdominis	*Starting position:* Lean on the exercise ball with bent forearms; upper body, buttocks and legs in a straight line, keep tension. *Execution:* Lower arms perform a circular movement with the ball, keep body tension.
#5 Rowing with Exercise Band in Standing Position	- M. latissimus dorsi - M. rhomboideus minor et major - M. trapezius pars ascendens, descendes und transversa	*Starting position:* Stand hip-width apart, knees slightly bent, upper body bent forward. Back is straight. Hold the exercise band around the feet with both hands, arms slightly bent. *Execution:* Draw shoulder blades together, then tighten the exercise band with bent arms, hold position briefly, then slowly return to starting position.
#6 Pelvic lift with Exercise Ball	- M. gluteus maximus - M. erector spinae pars lumbalis - M. deltoideus	*Starting position:* Supine, heels on the exercise ball, body forms a straight line and has tension, only shoulders and arms support the body, head lies relaxed on the floor in extension of the spine. *Execution:* Heels are pulled towards the buttocks, pelvis is lifted up to 90° flexion in the knee joint, then slowly return to starting position.
#7 Squat with Bar	- M. gluteus maximus - M. quadriceps femoris - M. erector spinae - M. deltoideus	*Starting position:* Feet slightly wider than shoulders, knees slightly out, hold broomstick above head with arms extended. *Execution:* Lower buttocks backwards in a controlled manner, back remains straight, do not push knees over toes, always turn knees slightly outwards, bend until 90° flexion in knee joint, then return to starting position.
#8 Neck Press	- M. longus colli - M. longus capitis - M. obliquus capitis	*Starting position:* Sitting on the floor, training band around the back of the head, holding both ends with the hands, upper body upright, back straight, hands at forehead level in front of the head. *Execution:* Head is moved backwards slowly and in a controlled manner.

**Table 4 jfmk-07-00088-t004:** Development of the posture parameters over time (mean ± standard deviation).

Group	Zoom		Control	
Time	Pre	Post	Pre	Post
Head protrusion [% BH]	4.4 ± 1.8	1.8 ± 1.3	4.0 ± 1.6	3.9 ± 1.5
Shoulder protrusion [% BH]	3.1 ± 1.8	1.4 ± 1.0	3.2 ± 1.5	3.1 ± 1.6
Hip anteversion [% BH]	3.1 ± 1.4	2.2 ± 1.2	3.1 ± 1.1	3.3 ± 4.7
Trunk tilt [°]	3.6 ± 1.8	2.1 ±1.3	2.8 ± 1.3	3.6 ± 1.4
Pelvic tilt [°]	9.5 ± 5.6	9.7 ± 3.6	9.9 ± 3.7	9.7 ± 4.4
Sacrum angle [°]	21.8 ± 7.9	17.9 ± 6.0	17.8 ± 5.8	12.4 ± 5.9

% BH = percentage of body height.

## Data Availability

The data is available for qualified requests.

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
