# Peer review of "Effects of Feedback-Supported Online Training during the Coronavirus Lockdown on Posture in Children and Adolescents"

_jfmk, 2022, doi:10.3390/jfmk7040088_

Round 1
Reviewer 1 Report
Title: Effects of feedback-supported online training during the Coronavirus lockdown on posture in children and adolescents
Manuscript ID: jfmk-1964792
This manuscript presented a research study to examine whether a six-week, feedback-supported online training programme (the British variant of the word program used in the manuscript) could improve posture parameters in young soccer players. Almost this manuscript is well written except the “1. Introduction” part, please see below my comments for the authors to consider and address:
1. Abstract, line 16 – “(2) Methods. Data of 170 adolescent soccer players (age 15.6 ± 1,6 years) was analyzed.” – term data is considered as plural, please modify this sentence as “ Data…..were analyzed”.
2. “1. Introduction” – this section is quite short, please try to modify it by explaining previous contributions clearly and giving more details. A quick suggestion, on the page 2, line 62 – write the research gap clearly with respect to the reported work in the paper.
3. Section 2.1 “Subjects”– please write more clearer about the participants group, the Zoom group and the control group, differentiate them more clearly to avoid any ambiguity. Also, same should be clarified in the “Abstract”.
4. Page1, Abstract, line 16 and Section 2.1 “Subjects”, “Table 1. Anthropometric data of the subjects” –decimal numbers are represented using a comma (,) which is, of course, a standard German practice, while rest of the paper used a decimal point (.), please take care of this.
5. Section 2.2 “Posture analysis” and other sections where corona measures are written in quite detail – if it is not really necessary to explain the corona precaution measures in the main text, please try to move it in the appendix section. It obstructs the readability of the experiment procedure related with the data collection. In the main text, writing – “appropriate corona precaution measures were followed strictly” – will be enough.
6. Page 3, Figure 1 and page 10, Figure 4 – for analyzing posture parameters, if some participant had deformation in ear, shoulder, or hip area, how does it affect the posture analysis? And, is this meaningful to generalize it from one participant to another?
7. In abstract, total number of the participants given as 170 (86+84), also same figures in the Table 1 but in the section 2.1 “Subjects” – “A total of 96 adolescents………… , out of a possible 115 subjects.” – doesn’t clearly explain this thing. Please consider to explain it in a better way.
Overall, this manuscript is well written but still lacks in the section “1. Introduction”; following other comments and suggestions can improve the quality of this paper.

Reviewer 2 Report
This article is well written. I think the author's efforts are reflected in the method and discussion. But I hope the author can enrich his conclusion.
Author Response
Dear Reviewer,
Thank you very much for the positive evaluation of our study. We have made a few improvements and, on your advice, added one more aspect to the conclusions (line 380 f).
Round 2
Reviewer 1 Report
Title: Effects of feedback-supported online training during the Coronavirus lockdown on posture in children and adolescents
Manuscript ID: jfmk-1964792
This review is in response to the manuscript revised as per the comments given in the first review on October 6, 2022. This revised manuscript has addressed and incorporated all comments and suggestions. Improving “1. Introduction” section and including further references enhanced the objectivity of the manuscript and justified the research gap clearly. Other comments and suggestions are also well responded in the revised manuscript.
I do not have additional comments besides please check for typos carefully, if any.
Good luck.
